# Non-invasive, opsin-free mid-infrared modulation activates cortical neurons and accelerates associative learning

Jianxiong Zhang [1,6], Yong He[2,6], Shanshan Liang[1,6], Xiang Liao [3,6], Tong Li[1], Zhi Qiao [2], Chao Chang [2✉], Hongbo Jia [4,5✉] & Xiaowei Chen [1✉]

Neurostimulant drugs or magnetic/electrical stimulation techniques can overcome attention deficits, but these drugs or techniques are weakly beneficial in boosting the learning capabilities of healthy subjects. Here, we report a stimulation technique, mid-infrared modulation (MIM), that delivers mid-infrared light energy through the opened skull or even non-invasively through a thinned intact skull and can activate brain neurons in vivo without introducing any exogenous gene. Using c-Fos immunohistochemistry, in vivo single-cell electrophysiology and two-photon $Ca^{2+}$ imaging in mice, we demonstrate that MIM significantly induces firing activities of neurons in the targeted cortical area. Moreover, mice that receive MIM targeting to the auditory cortex during an auditory associative learning task exhibit a faster learning speed (~50% faster) than control mice. Together, this non-invasive, opsin-free MIM technique is demonstrated with potential for modulating neuronal activity.

[1] Brain Research Center and State Key Laboratory of Trauma, Burns, and Combined Injury, Third Military Medical University, Chongqing, China. [2] Innovation Laboratory of Terahertz Biophysics, National Innovation Institute of Defense Technology, Beijing, China. [3] Center for Neurointelligence, School of Medicine, Chongqing University, Chongqing, China. [4] Advanced Institute for Brain and Intelligence and School of Physical Science and Technology, Guangxi University, Nanning, China. [5] Brain Research Instrument Innovation Center, Suzhou Institute of Biomedical Engineering and Technology, Chinese Academy of Sciences, Suzhou, China. [6] These authors contributed equally: Jianxiong Zhang, Yong He, Shanshan Liang, Xiang Liao. ✉email: changc@xjtu.edu.cn; jiahb@sibet.ac.cn; xiaowei_chen@tmmu.edu.cn

It has become increasingly popular for people to take some neurostimulant drugs as nootropics[1] that are expected to enhance cognition and learning[2] beyond the initially approved therapeutic purposes, such as curing attention deficits[3]. Likewise, brain stimulation techniques, including, e.g., transcranial magnetic stimulation and transcranial direct-current stimulation, have also been extensively practiced in patients with similar expectations[4–7]. However, while those drugs or stimulation techniques could help alleviating the relevant deficits, their resulting effects in the diseased conditions do not necessarily imply that healthy subjects could benefit from them in boosting the normal learning capabilities.

Learning is an intricate process that involves highly specific patterns of neuronal activation[8–10], and the neocortex is known to be functionally relevant for the associative learning process[11–13]. In the past decade, specific manipulations of neuronal activities have been achieved by using the optogenetic technique[14], advancing the understanding of learning and memory formation[15]. Although having become highly popular for animal experiments, the optogenetics technique shows little potential for applications in healthy humans due to the requirement of introducing exogenous genes in the brain. Here, we present a fundamentally different energy stimulation technique, mid-infrared modulation (MIM), which delivers mid-infrared (MIR) light energy through opened skull or non-invasively through thinned intact skull to the brain and can significantly elevate neuronal firing rates in the targeted brain region. Notably, MIM induces neuronal firing in complete absence of any exogeneous gene. As a striking example, we demonstrate that MIM application in the auditory cortex of healthy adult mice during a sound-licking associative learning task boosts learning speed by ~50%.

## Results and discussion

We used a pulsed quantum cascade laser as the MIR light source for MIM in this study (see "Methods" for details). A MIR fiber (core diameter 100 μm, numerical aperture 0.27) delivered the MIR to the target region of the mouse brain (Fig. 1a), in a manner superficially similar to fiber-based optogenetics technique[16]. However, two major features fundamentally distinguish MIM from optogenetics. First, no exogenous gene was introduced into the brain. Second, the wavelength of stimulation light was 5.6 μm, which is in the mid-infrared spectrum (3–50 μm, as defined by the ISO 20473 standard) and far beyond the visible (VIS, 0.38–0.78 μm) to near-infrared (NIR, 0.78–3 μm) spectrum used in optogenetics[16].

**MIM activates cortical neurons in vivo**. We delivered the MIR with the following parameters: average irradiation power $9 \pm 0.5$ mW ($n = 5$ measurements at the fiber tip), pulse width 300 ns, repetition rate 100 kHz, and irradiation duration 20 s (for detailed protocol and explanation of parameters, see "Methods"). In order to yield repeatable and comparable results, we first placed the fiber tip closely above the cortical surface following craniotomy. After MIR delivery we performed immunohistochemistry for c-Fos (with DAPI staining to identify cells), a widely-used molecular marker of neuronal activation[17]. C-Fos+ cells were found in a bullethead-shaped volume that was mostly within layer 2/3 of the cortex (Fig. 1b). A control experiment by co-labeling of c-Fos and NeuN (Fig. 1c) showed that 88.9/84.3–90.0% ($n = 11$ slices from 4 mice, median/1st–3rd quartile, same notation for all subsequent data) of c-Fos+ cells emerged after MIR irradiation were neurons. The axial and lateral extent of the c-Fos+ cell distribution was both ~400 μm and the half-width of distribution was ~150 μm (Fig. 1d). Within this zone (e.g., outlined in Fig. 1b),

the proportion of c-Fos+ cells among all cells (DAPI stained) was 27.8/24.1–32.1 % ($n = 21$ slices from 7 mice).

We next performed a new set of dose-dependent tests and found that the proportion of c-Fos+ cells positively correlated with the irradiation time in our tested range between 5 s and 60 s (Fig. 1e, left 4 columns, "5 s": 8.0/5.9–11.5 %, $n = 11$ slices from 4 mice; "10 s": 19.0/9.9–22.5%, $n = 9$ slices from 4 mice; "20 s": 23.3/19.8–31.5%, $n = 12$ slices from 4 mice; "60 s": 64.0/51.3–74.0%, $n = 13$ slices from 4 mice). Having shown that MIM application through opened skull could reliably induce neuronal activation, we applied MIM to the mice with thinned intact skull (thickness remaining: 44/37–48 μm; $n = 12$ measurements). MIM through thinned intact skull (irradiation time of 20 s) also induced neuronal activation (Fig. 1e, "Thinned 20 s", c-Fos cell proportion: 12.3/9.5–21.0 %, $n = 8$ slices from 4 mice), albeit with a lower efficacy than that through opened skull ($P = 0.012$ comparing to "Opened 20 s", two-sided Wilcoxon rank-sum test).

We conducted a new set of experiments wherein MIR and VIS light were delivered to each side of the cortex of individual mice respectively (both through opened skull; Fig. 1f, and "Methods"), each via a fiber with the same geometric parameters and with the same irradiation power (~9 mW) and time (20 s). Copious c-Fos+ cells were present in the MIR- but not the VIS-targeted regions (Fig. 1g). A few c-Fos+ cells were found at superficial locations within 70 μm from the cortical surface (i.e., layer 1); cells here are generally not considered to be pyramidal neurons[18]. Thus, we recalculated the c-Fos+ cell proportion for this experiment by excluding these labeled superficial cells, and we found that in the VIS region the value was near zero (Fig. 1h, MIR: 25.3/23.3–37.8%, $n = 8$ slices from 4 mice; VIS: 0/0–0.6%, $n = 13$ slices from 4 mice, $P = 1e{-}4$, two-sided Wilcoxon rank-sum test).

How does MIR light activate neurons without opsins? There have been studies on photostimulation or photomodulation using light in the near-infrared spectrum[7,19] (typically 1–3 μm wavelength), for which heat-related mechanisms could account for the activation of nerve cells[20,21]. To test this hypothesis, we measured the cortical tissue temperature under MIR or VIS light irradiation in vivo (Fig. 1i and Supplementary Fig. 1) with the same configuration as that described above for the c-Fos experiment. Temperature elevation at a measuring site closest to the fiber tip in both conditions were less than 2 °C, though the VIS light-induced temperature elevation was even higher than that induced by MIR light (Fig. 1i; VIS light: 1.35/1.23–1.40 °C, $n = 7$ mice; MIR light: 0.62/0.51–0.69 °C, $n = 7$ mice, $P = 6e{-}4$, two-sided Wilcoxon rank-sum test), consistent with previous literature on heating effect of fiber-delivered light in brain tissues in vivo[22]. Accordingly, other previous experimental studies have shown that moderate heating (up to 2 °C) by VIS light either does not change or even suppresses neuronal activities in different brain regions, through activation of an inwardly rectifying potassium conductance[23,24]. Therefore, our results do not favor the hypothesis that general tissue heating could account for the MIR-induced neuronal activities under our in vivo experimental condition. However, our temperature measurement had a limited spatial resolution of ~100 μm. Thus, an alternative interpretation is that local temperature fluctuations probably occurring at a scale of a few micrometers or even nanometers could activate heat-sensitive ion channels and induce neuronal firing[21].

To demonstrate that MIM could indeed induce neuronal spike firing, we performed in vivo single-cell loose-patch recording in cortical neurons throughout a course of MIM application (Fig. 1j, k). For each recorded neuron (Fig. 1l), we counted the number of spikes per 20-s time window before, during and after MIM application time window (Fig. 1m, trial-averaged). The population data analysis showed that spike counts increased by ~36% in the MIM window ("pre": 2.2/0.7– 2.5, "MIM": 3.0/2.2–6.6, "post":

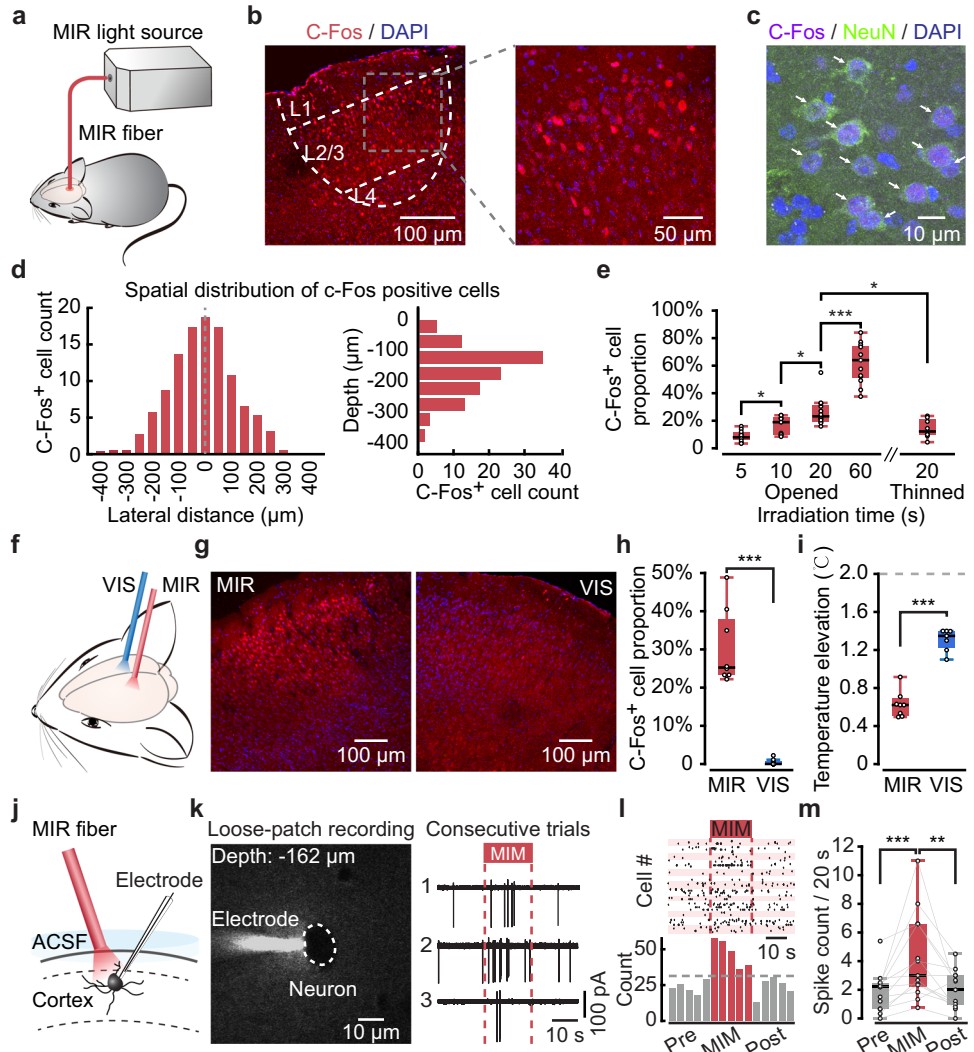

**Fig. 1 MIM application to the mouse cortex induces neuronal activities. a** General scheme for targeted MIR delivery to mouse brain in vivo. **b** Left image: a confocal image of a postmortem slice near the MIR target spot. Right image: magnified view of the left image (gray dashed box). White dashed lines: outlining the cortical layers and the central zone of c-Fos+ expression, a commonly used marker of neuronal activation. **c** Control experiment with co-labeling of c-Fos and NeuN. White arrows: cells positive with both c-Fos and NeuN. **d** Histograms of the lateral (left) and transverse (right) distributions of c-Fos+ cells; data points are pooled from 7 mice. See Methods for definition of cell counts. **e** C-Fos+ cell count in slice samples taken from different groups of mice ($n = 4$) performed with different MIR irradiation time and conditions (through opened skull or thinned intact skull). $P = 0.012$ (Opened 5 s versus 10 s), $P = 0.039$ (Opened 10 s versus 20 s), $P = 8.0054e−05$ (Opened 20 s versus 60 s), $P = 0.0121$ (Opened 20 s versus Thinned 20 s), two-sided Wilcoxon rank-sum test, n.s. $P > 0.05$, *$P < 0.05$, **$P < 0.01$, ***$P < 0.001$, applies for all panels in this figure. **f** Schematic showing the MIR-VIS control experimental design. **g** Fluorescence images of slices taken at the MIR target spot (left) or the VIS target spot (right), both images taken from the same animal. **h** Boxplots showing the c-Fos+ cell count in the MIR- or VIS-exposed regions taken from the same 4 mice (8 slices taken for the MIR group, 13 slices taken for the VIS group, $P = 1.0157e−04$, two-sided Wilcoxon rank-sum test). **i** Local cortical tissue temperature elevation ($n = 7$ mice) at the site closest to the fiber tip, measured by a miniature thermocouple probe under MIR or VIS irradiation. Dashed line indicates 2 °C, below which light stimulation has previously been shown to not change or even suppress neuronal activities ($P = 5.8275e−04$, two-sided Wilcoxon rank-sum test). **j** Schematic for single-cell loose-patch recording simultaneously with MIM application in vivo. **k** Left image: the patch electrode and target cell under live two-photon imaging navigation in vivo. Right traces: three consecutive trials of loose-patch recording from one example neuron. **l** Upper raster plot: pooled single-trial spiking data from all 13 recorded neurons. Lower histogram: summary of spike count. **m** Boxplots summarizing 13 recorded neurons, showing spike count before ("Pre"), during (MIM) and after ("Post") MIM application. $P = 2.4414e−04$ (Pre versus MIM), $P = 0.0054$ (MIM versus Post), two-sided Wilcoxon signed-rank test. The box-and-whisker plots indicate the median (central mark), 25th and 75th percentiles (bounds of box: Q1 and Q3), interquartile range (IQR: Q3–Q1), and the whiskers extending to the minima and maxima without considering outliers.

2.0/1.0–3.0, $n = 13$ cells, $P = 3e−4$ for "pre" versus "MIM", $P = 0.007$ for "MIM" versus "post", two-sided Wilcoxon signed-rank test). Note that, when viewed on a single-trial basis, only a fraction of recorded neurons exhibited a statistically significant elevation in firing in repeated MIM application windows (2 out of 13 cells that satisfied *$P < 0.05$ in two-sided Wilcoxon signed-rank test individually; Supplementary Fig. 2). In some of the

loose-patch experiments ($n = 6$ cells) we used a second micro-pipette for tetrodotoxin (TTX) delivery to verify that the recorded spiking waveforms were neuronal firing signals but not photo-electrical artefacts (Supplementary Fig. 3). Besides, we did not observe a change in spike waveform width over the course of MIM application (Supplementary Fig. 4). These experiments above together establish a basic proof-of-principle for MIM

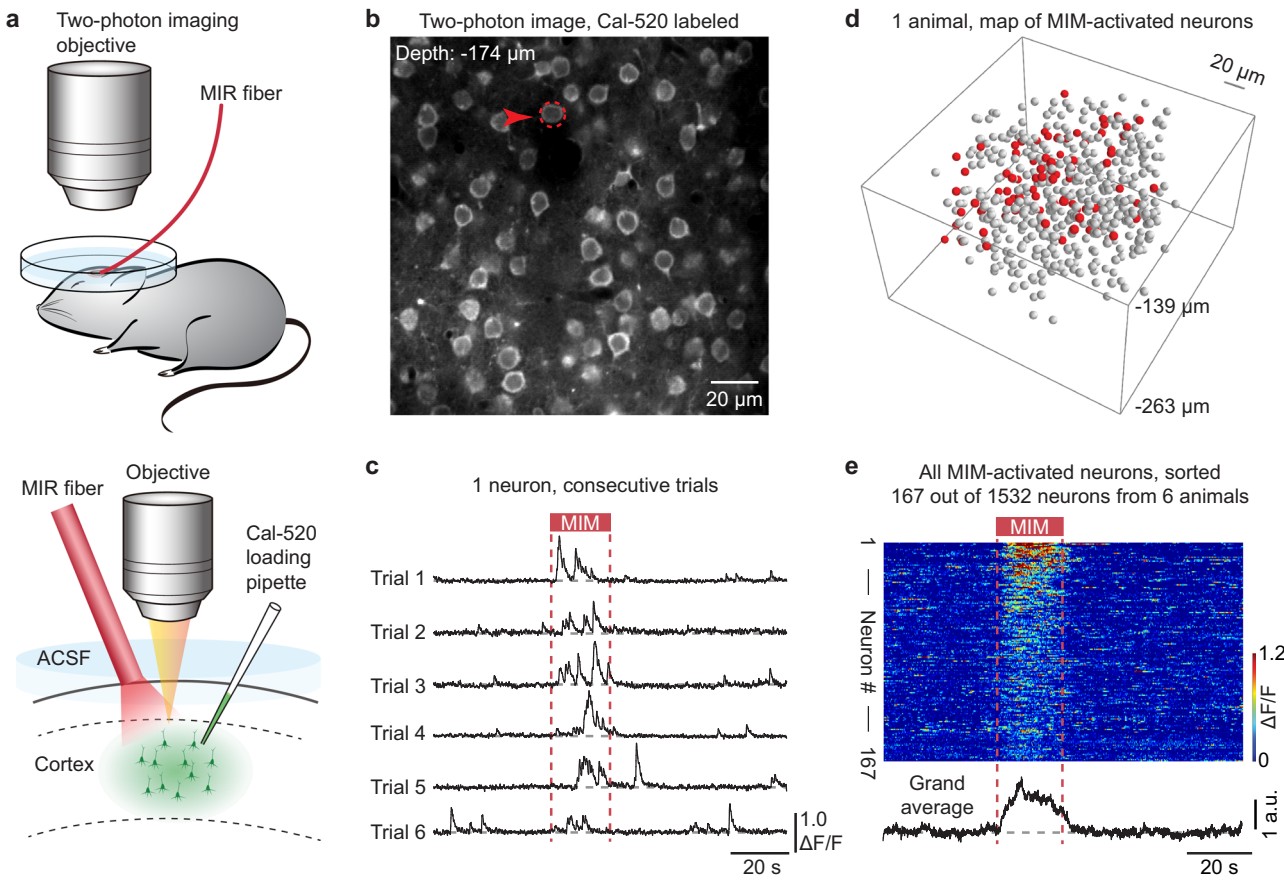

**Fig. 2 Spatiotemporal mapping of MIM-activated cortical neurons in vivo. a** Upper: cartoon showing the general scheme for the in vivo two-photon imaging experiments. Lower: cartoon showing the configuration of the MIR fiber tip, the two-photon imaging objective, and the Ca$^{2+}$ dye loading micropipette. ACSF: artificial cerebrospinal fluid. **b** An example in vivo two-photon image (averaged 100 frames). The red arrow and the dashed circle indicate the neuron for which Ca$^{2+}$ activity traces are shown in the next panel. **c** Ca$^{2+}$ activity traces of an example neuron (marked in **b**) in six consecutive trials. The two red dashed lines indicate the start and the end of the MIM application. **d** A reconstructed 3D map showing the positions of all MIM-activated neurons (red balls) and unaffected neurons (gray balls) in the imaged volume, consisting of eight focal planes from one example mouse. **e** Upper: a pseudo-colored plot summarizing the trial-integrated Ca$^{2+}$ activity trace for each neuron identified as MIM-activated. Neurons were sorted by their relative increment of activity level (from pre-MIM to MIM). Lower: a grand average of the Ca$^{2+}$ activity traces for all 167 MIM-activated neurons.

in vivo, demonstrating that MIR energy delivery indeed enhances neuronal spike firing in absence of any exogenous gene.

**MIM activates ~10% of cortical neuronal population in vivo.**
We next employed two-photon Ca$^{2+}$ imaging[25–27] to visualize live neuronal population activities during MIM application (Fig. 2a). Auditory cortical neurons were labeled with a Ca$^{2+}$-sensitive fluorescent dye, Cal-520 (e.g., Fig. 2b). We performed real-time (40 Hz) two-photon imaging over the entire course of MIM application (40 s before, 20 s MIR irradiation, 60 s after). An example neuron (Fig. 2c) showed that MIM application reliably induced Ca$^{2+}$ transients during the application time window over repeated trials. For each animal, we sequentially performed real-time imaging recordings (as the example above) at multiple cortical depths, which enabled mapping of the MIM-activated neurons (e.g., Fig. 2d). Note that a neuron was defined as MIM-activated if its trial-averaged Ca$^{2+}$ activity level during the MIM window exceeded the activity level detected for the baseline window (before the irradiation window) by more than 2-fold (see "Methods").

Out of a total of 1532 neurons from 6 mice, we found 167 MIM-activated neurons (Fig. 2e, for the complete dataset, see Supplementary Fig. 5). The activity level for the MIM-activated neurons recovered after the MIM window, returning to the same

level as before irradiation (see also the average Ca$^{2+}$ signal trace in Fig. 2e). Note that the proportion of MIM-activated neurons detected in these two-photon Ca$^{2+}$ imaging experiments was 10.2/4.0–17.5% ($n = 26$ imaging focal planes from 6 mice), which was lower than that in the c-Fos imaging results (27.8/24.1–32.1%, $n = 21$ slices from 7 mice). We speculate that this difference could result from methodological differences, i.e., c-Fos imaging captures the cumulative neuronal activities from a prolonged time (the time required for c-Fos expression[17] is much longer than the neuronal activation time), whereas two-photon Ca$^{2+}$ imaging resolves specific neuronal activities in real time. Nevertheless, our results from c-Fos imaging, two-photon Ca$^{2+}$ imaging, and single-cell loose-patch recording are highly consistent in demonstrating that MIM activates a substantial number of neurons in the targeted cortical region.

**MIM accelerates learning during an associative training task.**
Previous studies have reported that the activities of auditory cortical neurons are involved in auditory associative learning for mice[11,12,28,29]. We, therefore, tested whether and how MIM application in the auditory cortex might affect the learning course for a sound-licking associative learning task[26,29] (Fig. 3a). We recruited three cohort groups of naïve, healthy mice (8–9 weeks old at the beginning of training): the "control" group ($n = 13$

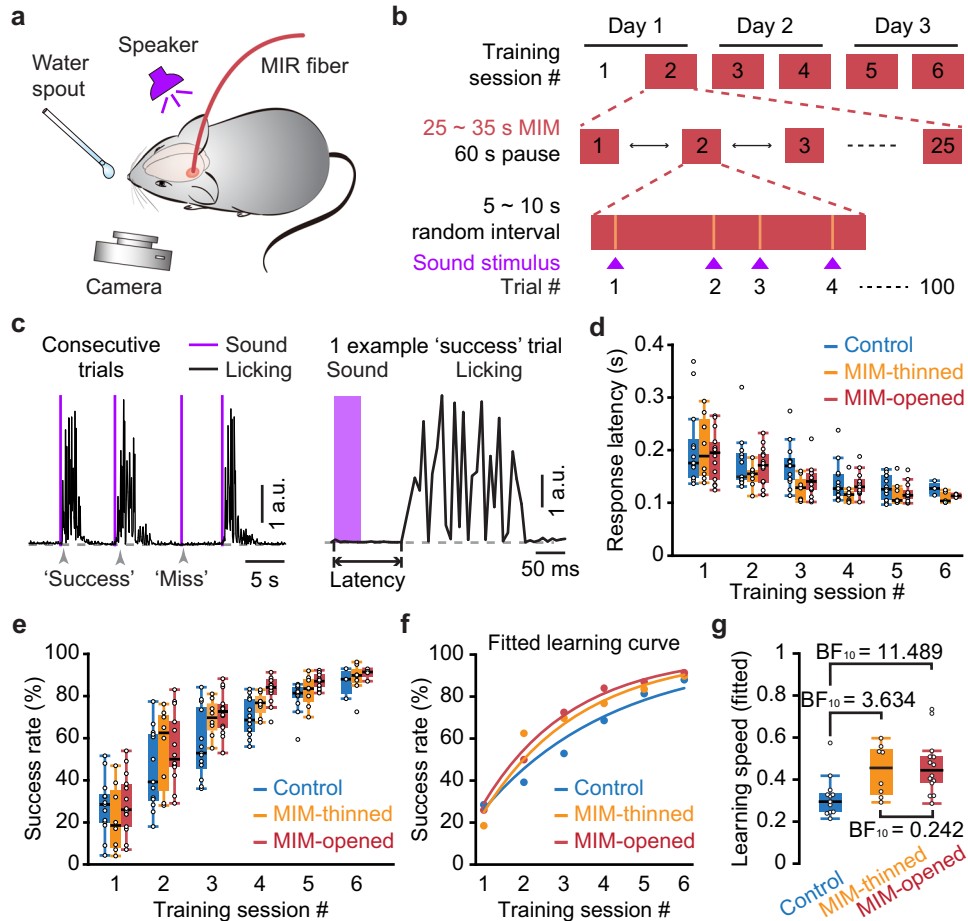

**Fig. 3 MIM in the auditory cortex accelerates learning during a sound-licking associative training task. a** Schematic illustrating the general scheme for the experiments using MIM in a sound-water associative training task. **b** Detail of the experimental protocol for MIM application throughout the training sessions. **c** Left trace, an example behavior recording consisting of four consecutive trial events, in which the third event was a "miss" and the other three events were "success". Right trace, enlarged view of one successful licking response event to illustrate how the response latency is determined. **d** Boxplots showing the latency of licking response for each group in each training session. "Control": mice with no treatments ($n = 13$ mice), "MIM-opened": mice with MIM applied through a small craniotomy ($n = 14$ mice), "MIM-thinned": mice with MIM applied through thinned intact skull ($n = 8$ mice). **e** Boxplots showing data for each training session and each group, representing the success rate for sound-evoked licking response. **f** Fitted learning curve for each group, showing the data averaged across the animals in each group respectively. **g** Boxplots showing the learning speed inferred from the fitted learning curve. Each data point represents one animal whose learning curve was individually fitted. Values marked on group comparison links: Bayes factor $BF_{10}$ (Bayes factor hypothesis testing, see "Methods" for details). $BF_{10} = 0.242$ (evidence of absence), $BF_{10} = 3.634$ (evidence of presence), $BF_{10} = 11.489$ (strong evidence of presence). The box-and-whisker plots indicate the median (central mark), 25th and 75th percentiles (bounds of box: Q1 and Q3), interquartile range (IQR: Q3-Q1), and the whiskers extending to the minima and maxima without considering outliers.

mice) that did not receive any MIM application, the "MIM-opened" group ($n = 14$ mice) that received MIM application through opened skull (a small craniotomy of ~200 μm diameter over the auditory cortex) and the "MIM-thinned" group ($n = 8$ mice) that received MIM application through thinned intact skull. All groups went through the same training program (consisting of 6 sessions). From session #2 on, animals in the two MIM groups received MIM application during the task engagement time windows (Fig. 3b, 25–35 s per engagement window with a 60 s pause between each). A typical example of a behavioral recording over consecutive trials during the learning process is shown in Fig. 3c, in which there were three successful response events (licking initiated within 0.5 s after sound stimulus onset) and one missed event (either no licking or licking initiated later than 0.5 s after sound).

For each animal at each training session, we calculated the average behavioral response latency of successful trials (Fig. 3d) as well as the overall response success rate per session (Fig. 3e). We applied the Bayes factor hypothesis testing[30] to test whether MIM

application induced a significant effect, as reported by the two-sided Bayes factor $BF_{10}$ ("Methods"). At the starting session (session #1), there were no obvious differences between any two groups in either the response latency ('control': 0.175/0.146–0.220 s; "MIM-thinned": 0.188/0.150–0.257 s, $BF_{10} = 0.399$ (absence of evidence); "MIM-opened": 0.194/0.144–0.215 s, $BF_{10} = 0.413$ (absence of evidence); "MIM-thinned" versus "MIM-opened": $BF_{10} = 0.445$ (absence of evidence)) or response success rate ("control": 28.5/19.7–33.4 %; "MIM-thinned": 18.5/8.2–35.4 %, $BF_{10} = 0.518$ (absence of evidence); "MIM-opened": 26.0/17.9–38.2%, $BF_{10} = 0.365$ (absence of evidence); "MIM-thinned" versus "MIM-opened": $BF_{10} = 0.579$ (absence of evidence).

At the end of the training procedure (session #6), there was slightly more evidence, yet not significant, that MIM groups exhibited a different response time than control group ("control": 0.123/0.117–0.138 s; "MIM-thinned": 0.103/0.100–0.120 s, $BF_{10} = 2.081$ (absence of evidence); "MIM-opened": 0.113/0.110–0.116 s, $BF_{10} = 1.375$ (absence of evidence)). Similarly, in session

#6 neither MIM group exhibited a different success rate than control group ("control": 88.0/81.1 –91.8 %; "MIM-thinned": 89.8/86.3–92.5%, $BF_{10} = 0.528$ (absence of evidence); "MIM-opened": 91.4/89.0–92.3%, $BF_{10} = 0.740$ (absence of evidence)). This result is not surprising because all three groups of animals were healthy and recruited without any initial bias, thus they could reach the same level of performance by the end of training procedure. However, when considering the entire training procedure (sessions #2–#6 when MIM was applied) as one test subject, both MIM groups exhibited a significantly higher response success rate than the control group ('control' versus "MIM-thinned": $BF_{10} = 5.663$ (evidence of presence), "control" versus 'MIM-opened': $BF_{10} = 18.262$ (strong evidence of presence)), and there was no difference between the two MIM groups ("MIM-thinned" versus "MIM-opened": $BF_{10} = 0.219$ (evidence of absence)).

To verify this result, we fitted a learning curve for each mouse (Fig. 3f shows the group average, for protocol see "Methods") using an exponential function[31], and then defined the learning speed by the fitted exponent factor. Comparing to the "control" group, MIM boosted learning speed by ~50% for either group with thinned intact skull or opened skull (Fig. 3g; fitted exponent factor, "control": 0.29/0.25–0.33; "MIM-thinned": 0.45/0.32–0.54, $BF_{10} = 3.634$ (evidence of presence); "MIM-opened": 0.44/0.38–0.51, $BF_{10} = 11.489$ (strong evidence of presence)). Note that, in this set of associative learning experiments, we applied MIM repeatedly over multiple trials (Fig. 3b). Thus, the reduced neuronal activation efficiency through thinned intact skull obtained by the c-Fos imaging experiments with a single-shot MIM (Fig. 1d) is not incompatible with the result here that, MIM through thinned intact skull achieved nearly the same degree of learning acceleration as that of MIM through opened skull ("MIM-thinned" versus "MIM-opened", $BF_{10} = 0.242$ (evidence of absence)).

Associative learning is generally assumed to require synaptic plasticity, which in mammals is known to depend on the temporal coincidence of synaptic and somatic activities[32–34]. Thus, we speculate that the observed acceleration of sensory-behavior associative learning likely results from extra MIM-induced somatic firing activities that coincide temporally with the task-relevant synaptic input activities occurring during the engagement window. In this scenario, the increase in coincident synaptic and somatic activities would result in an overall increase in the amount of task-relevant synaptic plasticity during a given task engagement duration, thus accelerating the learning. From another viewpoint[35], the MIM-induced activation of neurons, could promote the recruitment of these neurons into an engram cell population[36], thus accelerating the learning.

In this study, we demonstrate MIM, a non-invasive and opsin-free neuronal stimulation technique that is fundamentally different from the well-known and widely-deployed optogenetics technique[14,16]. MIM reliably enhances firing activities in a fraction of neuron within a precisely targeted and spatially confined volume of brain tissue in vivo, in complete absence of exogenous gene. Like that optogenetics application is capable of artificially alter memory, sensory perception or behavior control[37,38], we show here that MIM application in the auditory cortex profoundly accelerates the sound-licking associative learning (Fig. 3). Together, our results illustrate the utility of this promising technique for enhancing brain learning functions with mid-infrared light energy.

## Methods

**Animals**. C57BL/6J male mice (2–3 months old) were provided by the Laboratory Animal Center of the Third Military Medical University. The mice were housed in a temperature- and humidity-controlled room on a cycle of 12 h light/dark (lights

off at 19:00). All experimental procedures were performed in accordance with institutional animal welfare guidelines with the approval of the Third Military Medical University Animal Care and Use Committee.

**Mid-infrared light source**. A quantum cascade mid-infrared (MIR) laser (Daylight solution Inc., Model "MIRcat") was used for this study. The laser contains four modules for continuous wavelength tuning but it was set to work constantly at 5.6 μm wavelength for this study. The output was coupled with a MIR fiber (IRF-Se-100, Label No. 11822-01) with a core diameter of 100 μm and a numerical aperture (NA) of 0.27. For all experiments in this study, the average power at the fiber tip (measured in the air) was at a stable level of $9 \pm 0.5$ mW with the following configurations: pulse duration 300 ns, repetition rate 100 kHz (equivalent to 300 mW peak power at a duty cycle of 3%). This MIR pulsing laser was initially designed for the purpose of infrared spectroscopy; when we used it for this study, all the parameters were configured to maximize the total average output power at the 5.6 μm wavelength (spectral-wise as well, the laser delivers maximum output power at this wavelength).

**Visible light source**. As a control for the c-Fos imaging experiment (Fig. 1), we used standard mono-wavelength visible (VIS) lasers with an optical fiber of the same geometry as that of the MIR fiber. The laser models were MBL-III-473-50mW and MBL-III-594-100mW (Changchun New Industries Optoelectronics Technology Co., Ltd, China) with working wavelengths 473 nm and 594 nm, respectively. These VIS lasers were delivered by an optical fiber that had the same geometric parameters as that for MIR light (core diameter 100 μm, NA = 0.27). The output power at fiber tip was adjusted and exposure time was set to be the same as those of the MIR light (9 mW, 20 s). For c-Fos imaging experiments, we did not find a significant difference between the results obtained by using the two different visible wavelengths, and thus these data were pooled together. For temperature measuring, we used only the 594 nm one as the VIS light source.

**Immunohistochemistry**. For the c-Fos experiments (Fig. 1), the animal was first head-fixed under general anesthesia with isoflurane (1.5%). The fiber tip was placed either above the brain surface via an open craniotomy or over the thinned intact skull (Fig. 1e). After the MIM application, the animal was kept under anesthesia for 90 min for c-Fos expression. The animal was then transcardially perfused with 4% paraformaldehyde (PFA) and the brain was fixed in 15% sucrose in 4% PFA and refrigerated overnight at 4 °C. Coronal sections (thickness: 44 μm) encompassing the fiber target spot were made by a vibratome. The c-Fos immunostaining was performed with the primary antibody: Anti-c-Fos (ABE457, Millipore rabbit polyclonal antibody, Lot# 3221531, 1:500 dilution), followed by the secondary antibody: Alexa Fluor 594 goat anti-rabbit (A11012, Invitrogen, Lot# 2119134, 1:500 dilution). DAPI (4′,6-diamidino-2-phenylindole, D9564, Sigma-Aldrich, 1:1000 dilution) was used to stain cell nuclei for all c-Fos+ cell counting experiments (Fig. 1). In an independent set of control experiments (Fig. 1c), antibody Anti-NeuN (MAB377, Millipore, mouse polyclonal antibody, 1:200 dilution) was used to label neurons, followed by the secondary antibody: Alexa Fluor 488 donkey anti-mouse (A21202, Invitrogen, Lot# 1644644, 1:500 dilution). Sections were mounted on slides with coverslips and imaged using a scanning confocal microscope (TCS SP5, Leica). Cells exhibiting both c-Fos and DAPI were defined as c-Fos+ cells, and cells exhibiting DAPI alone were defined as c-Fos negative cells. In the control experiments with c-Fos and NeuN co-labeling, cells exhibiting NeuN were defined as neurons.

**Cortical tissue temperature measurement**. We performed the tissue temperature measurement protocols[22,23] with an ADINSTRUMENT acquisition system (PowerLab 4/35) coupled to a T-type hypodermic thermocouple (MT 29/5, Physitemp). Animal was first head-fixed under general anesthesia with isoflurane (1.5%) and the MIR (or VIS in an independent set of control measurements) fiber was placed above cortical surface in the same configuration as in the c-Fos imaging experiments. The brain surface was immersed with ACSF solution that was circulated and controlled at 37 °C. The thermo sensor was inserted into the cortex from a horizontal orientation (see Supplementary Fig. 1a) by a fine micromanipulator to adjust the depth and lateral position of the sensor tip. Dura mater under the craniotomy was removed for these measurements in order to expose the cortical tissue with a small bulge, allowing the sensor to be inserted from horizontal orientation. We expected that the actual cortical tissue temperature change with intact dura in the other physiological experiments (c-Fos, loose-patch, two-photon imaging) would be less than what we measured in absence of dura. From the onset time of MIR or VIS light irradiation, the sensor readout reached a plateau in less than 10 s and we took this plateau value as the measured value.

**Loose-patch recording and TTX local application**. For loose-patch recordings in cortical neurons in vivo, we used the "shadow-patching" procedure[39–42], except that we did not rupture the membrane of targeted cells to maintain a loose-patch configuration. Cell-attached recordings were performed with an EPC10 amplifier (HEKA Elektronik, Germany). The glass electrode was filled with normal ACSF (with ~10 μM OGB1-6K dye to be visualized under two-photon microscope) and had a tip resistance of 5–8 MΩ. The electrode penetrated the dura with positive

pressure of ~100 mbar and was then reduced to ~ 20–30 mbar to obtain good images of both electrode and shadow neurons under two-photon imaging. Raw signals were filtered at 10 kHz and sampled at 20 kHz using Patchmaster software (HEKA Elektronik, Germany). In some experiments (6 neurons), a second micropipette (the tip resistance of 5–8 MΩ) was inserted to be close to the recorded neuron (see Supplementary Fig. 3), containing tetrodotoxin (TTX, 10 μM) which was injected for 10 s at a pressure of 50 mbar.

**Two-photon Ca$^{2+}$ imaging**. For acute two-photon Ca$^{2+}$ imaging experiments (Fig. 2), we exposed the right auditory cortex of the mouse[27,40]. In brief, the animal was anaesthetized by isoflurane and kept on a warm plate (37.5 °C). The skin and muscles over the Au1 were removed after local lidocaine injection. A custom-made plastic chamber was glued to the skull with cyanoacrylate glue (UHU), followed by a small craniotomy (~2 mm × 2 mm) (the centre point: Bregma −3.0 mm, 4.5 mm lateral to midline). These stereotaxic coordinates correspond to a larger region of the auditory cortex, including the primary auditory cortex (Au1), the dorsal and ventral secondary auditory cortex (AuD and AuV) as well as a part of the adjacent temporal association cortex (TeA)[11]. After performing the craniotomy, the animal was transferred to the imaging rig with a head-fixation chamber. The chamber was perfused with normal artificial cerebral spinal fluid (ACSF) containing 125 mM NaCl, 4.5 mM KCl, 26 mM NaHCO$_3$, 1.25 mM NaH$_2$PO$_4$, 2 mM CaCl$_2$, 1 mM MgCl$_2$ and 20 mM glucose (pH 7.4 when bubbled with 95% oxygen and 5% CO$_2$). The Ca$^{2+}$ dye, Cal-520 AM (AAT Bioquest), was dissolved in DMSO with 20% Pluronic F-127 to a final concentration of 567 μM for bolus loading. The loading procedure was conducted to label L2/3 neurons in Au1[25,26,43]. The pipette containing dye was inserted into the cortex up to 200 μm deep from pial surface. About 2 h after dye injection to allow sufficient cellular uptake, two-photon imaging was performed with a custom-built two-photon microscope system based on a 12.0 kHz resonant scanner (model "LotosScan 1.0", Suzhou Institute of Biomedical Engineering and Technology[44,45]). Two-photon excitation light was delivered by a mode-locked Ti:Sa laser (model "Mai-Tai DeepSee", Spectra Physics) at a near-infrared (NIR) wavelength of 920 nm. A 40×/0.8 NA (Nikon) water-immersion objective was used for imaging, and this objective had a long working distance of 3.5 mm and a large access angle of ~40 degree to allow micropipette manipulations from the side. We placed the fiber tip from an oblique angle to the cortical surface and protected it by a glass micropipette to avoid excess contact with ACSF. The typical size of the imaging field-of-view (FOV) was ~200 μm × 200 μm. In a typical experiment, time-lapse imaging recording at different focal depths could be performed sequentially. The average power of the output laser (under the objective) was in the range of 30–120 mW, depending on the depth of imaging.

**Associative training**. For the sound-licking associative training experiments (Fig. 3) we adapted the training protocol from our previous studies[26,28,40]. Sound stimuli were delivered by an ED1 electrostatic speaker driver and a free-field ES1 speaker (both from Tucker Davis Technologies). The distance from the speaker to the mouse ear (contralateral to the imaged A1) was ~6 cm. The sound stimulus was produced by a custom-written, LabVIEW-based program (LabVIEW 2012, National Instruments) and transformed to analog voltage through a PCI6731 card (National Instruments). Sound levels tested with a microphone placed ~6 cm away from the speaker were calibrated by a pre-polarized condenser microphone (377A01 microphone, PCB Piezotronics Inc.). For broadband noise (BBN, bandwidth 0–50 kHz), the sound level was ~65 dB sound pressure level (SPL). A waveform segment of BBN was first generated, and the same waveform segment was used for all experiments (i.e., a "frozen noise"). The duration of a sound stimulus was 50 ms.

Before training, the animal was implanted with the headpost under isoflurane anaesthesia and then allowed to recover for 5 days. After one night (20:00–08:00) of water restriction, the mouse was habituated for head fixation for 2–3 days. During habituation, the mouse received water in the behavioral setup exclusively. For animals in "MIM-opened" or "MIM-thinned" group, the fiber tip was fixed above the cortical surface or the thinned section of skull, respectively, before the beginning of each session from session #2 on (and was removed after the end of session). The stereotaxic coordinates used for the placement of the fiber tip were the same as those for two-photon Ca$^{2+}$ imaging experiments in the auditory cortex. Careful management of water consumption was taken to ensure that animals in all groups received similar volume of water and had a similar level of motivation throughout all sessions and days. During the training sessions, the animal was head-fixed to the training rig. A droplet of water was formed at a spout by automatically controlled pumping (pumping duration, 20 ms) at 100 ms after the end of the sound stimulus (in total, 50 + 100 = 150 ms from the stimulus onset). Water droplets remained at the spout after being delivered so that the animal could always obtain water if ever it voluntarily made a licking action at any time after water was delivered. If the animal had not licked before the next trial occurred, a new droplet would replace the previous one at the spout. The spout was positioned at a distance of approximately 3–4 mm from the animal mouth (and with no visible ambient light) such that the animal had to voluntarily stretch out its tongue to probe and acquire water droplets on the spout. Licking actions were monitored by a camera (frame rate 30 Hz) under NIR illumination that was invisible to the animals. There was no cue, stimulus or rewarding/punishing object

beyond the sound and the water. We did not apply any punishment for incorrect licking timing.

One training session contained 100 sound stimulation events. These stimuli were delivered in 25 discrete engagement time windows, each consisting of four events with random inter-trial intervals (in the range of 5–10 s, longer than the duration of a licking action). The rationale of using a randomized inter-trial interval setting was to avoid the possible effect of the rhythmic predicative responses that have been known to exist in mice[28]. Each engagement time window was followed by a 60-s pause with neither sound nor water being present. For animals in MIM group, the MIR irradiation was turned on (by a sound-free electronic shutter) during the engagement time window (which was 20–35 s depending on the random inter-trial interval), similar to the MIM applications in the two-photon imaging experiments.

A success sound-evoked licking event was defined as an event in which the animal initiated a licking action within 500 ms from sound stimulus onset, otherwise the event was defined as a miss. For a session, the success rate was defined as the number of success events divided by the total number of events. For analysing the licking response latency, the behavior monitoring videos were first inspected by humans and the video frame of licking action onset was marked for each trial. The response latency was then defined as the duration from the sound stimulation onset to the first video frame in which the animal stretched out its tongue (as shown in Fig. 3c right).

**Ca$^{2+}$ imaging data analysis**. We analyzed our data using custom-written software in LabVIEW 2012 (National Instruments), Igor Pro 5.0 (Wavemetrics), and MATLAB 2014a (MathWorks)[46]. To correct motion-related artefacts in imaging data, we used a frame-by-frame alignment algorithm to minimize the sum of squared intensity differences between each frame image and a template, which is the average of the selected image frames.

To extract fluorescence signals, we visually identified neurons and performed the drawing of regions of interest (ROIs) based on fluorescence intensity. Fluorescence changes (f) were calculated by averaging the corresponding pixel values in each specified ROI. Relative fluorescence changes Δf/f = (f−f$_0$)/f$_0$ were calculated as Ca$^{2+}$ signals, where the baseline fluorescence f$_0$ was estimated as the 25th percentile of the entire fluorescence recording. Glial cells were excluded from our analysis according to morphology and Ca$^{2+}$ transient time course.

For the Ca$^{2+}$ signals from each neuron recorded in the two-photon imaging data, we performed automatic Ca$^{2+}$ transient detection based on thresholding criteria regarding peak amplitude and rising rate[28,47]. The noise level was set to 3 times the standard deviation of the baseline (window length: 1 s), the peak amplitude, and the rate of rise of Ca$^{2+}$ signal were calculated to determine it as a true transient or not. After that, the trace of the detected Ca$^{2+}$ transient was first extracted by exponential infinite impulse response (IIR) filtering (window length: 200 ms) and then subtracted from the original signal. The residual fluorescence trace was used as the baseline for a next transient detection, which is similar to the peeling approaches previously described by the Helmchen laboratory[48]. During the licking task, licking activities were semi-automatically tracked from the monitoring movie and quantified as a time course[26,28]. The tongue movements during licking were measured at a particular location (ROI) around the mouth of mouse, and the amplitudes of licking strength were calculated as ROI-based image intensity difference between the video frames.

**Learning curve fitting and definition of learning speed**. For each animal in the learning experiment (Fig. 3), a simple exponential function was used to fit to the licking success rate $y$ against the training session number $x$ as

$$y = 1 - a \cdot e^{b \cdot x} \tag{1}$$

where $a$ and $b$ are two fitting parameters, and parameter $b$ is defined as the learning speed. The fitting algorithm was standard least-square regression, performed in Matlab using its built-in library function "nlinfit".

**Statistics and reproducibility**. In the text, summarized data are presented as the median/25th–75th percentiles. In the figures, the box-and-whisker plots indicate the median (central mark), 25th and 75th percentiles (bounds of box: Q1 and Q3), interquartile range (IQR: Q3−Q1), and the whiskers extending to the minima and maxima without considering outliers. An observation is defined as an outlier if it falls more than 1.5 × IQR above the third quartile or below the first quartile. We used Matlab 2014a to generate the boxplots. To compare data between groups, we used non-parametric Wilcoxon rank-sum test (unpaired) and Wilcoxon signed-rank test (paired) to determine statistical significance between them. In addition, we also applied the Bayes factor hypothesis testing[30], including Bayesian t-test for single-session tests and Bayesian ANOVA for multi-session tests on the behavioral data (Fig. 3). We used the two-sided Bayes factor BF$_{10}$ and a criterium value $x = 3$ to define the outcome, as follows: BF$_{10}$ ≤ 1/3: evidence of absence (favoring null hypothesis H$_0$, no difference between groups); 1/3 < BF$_{10}$ ≤ 3: absence of evidence (data insufficient to support alterative hypothesis H$_1$); BF$_{10}$ > 3: evidence of presence (favoring H$_1$, corresponding to $P < 0.05$ in standard tests); BF$_{10}$ > 10: strong evidence of presence (strongly favoring H$_1$, corresponding to $P < 0.01$ in standard tests). Although the evidence that MIM application induced a positive effect was

inconclusive in sessions #1 and #6 in Fig. 3d and e, the Bayes factor favors the null hypothesis. For the Bayes factor hypothesis testing, we inherited the program from the original paper[30].

For the representative experiments in Fig. 1b, c, g, these experiments were repeated independently, and similar results were obtained from $n = 7$, 4, and 4 mice, respectively.

**Reporting summary**. Further information on research design is available in the Nature Research Reporting Summary linked to this article.

## Data availability
The data that support the findings of this study are available from the corresponding author upon reasonable request. Source data underlying Figs. 1–3 and Supplementary Figs. 1–5 are available as a Source data file. Source data are provided with this paper.

## Code availability
The codes supporting the current study have not been deposited in a public repository, but are available from the corresponding author upon request.

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

## Acknowledgements
The authors are grateful to Prof. Guozhi Liu for proposing the possible biological effects of mid-infrared light, Dr. Yousheng Shu and Dr. Bo Song for inspiring our work, to Ms. Jia Lou for help in composing and layout editing of the figures and to Dr. John H. Snyder for help in language editing the manuscript. This study was supported by grants from the National Natural Science Foundation of China (31925018, 31861143038, 31921003, 81671106) and the Chongqing Outstanding Young Investigator Fund Project (cstc2019jcyjjqx0001). X.C. is a member of the CAS Center for Excellence in Brain Science and Intelligence Technology, Shanghai Institutes for Biological Sciences, Chinese Academy of Sciences, and also a member of the Institute of Brain and Intelligence at the Third Military Medical University. C.C. acknowledges the support from the XPLORER PRIZE.

## Author contributions

C.C., H.J., and X.C. conceived the project. J.Z. and X.C. designed the experiments; J.Z., Y.H., and Z.Q. performed the experiments; J.Z., S.L., Y.H., X.L., T.L., X.C., and H.J. performed the data analysis; C.C., H.J., and X.C. inspected the data and evaluated the findings; H.J. and X.C. wrote the manuscript with the help from all authors.

## Competing interests

The authors declare no competing interests.
