## [Peer Review File · Nature Communications]

Reviewer #1 (Remarks to the Author):

This paper uses a pulsed 5.6 μm laser to carry out optical stimulation in the mouse brain and in patch-clamped neurons.

1 The authors make a big deal of their study being different from traditional optogenetics approaches requiring the introduction of a light sensitive opsin into the neuronal cells. However in my opinion they mostly neglect a large body of work called "infrared neural stimulation". There is a good review of the subject

Chernov M, Roe AW. Infrared neural stimulation: a new stimulation tool for central nervous system applications. *Neurophotonics*. 2014;1(1):011011. doi:10.1117/1.NPh.1.1.011011

2 The wavelength they use (5.6 μm) may be novel but other wavelengths 980 nm, 1.45 μm , 1.87 μm , 2.12 μm have all been used. Absorption by water is what they all have in common and a small spot and a pulsed mode provide a confined thermal gradient to activate heat sensitive TRP ion channels

3 In fact TRPV4 was shown to be responsible in one study with 1.87 μm . Albert ES, Bec JM, Desmadryl G, et al. TRPV4 channels mediate the infrared laser-evoked response in sensory neurons. *J Neurophysiol*. 2012;107(12):3227-3234. doi:10.1152/jn.00424.2011

4 In addition to stating the average power of the laser (9mW) they should also state the peak power (300 mW)

Michael R Hamblin

Reviewer #2 (Remarks to the Author):

The authors show that optical pulsatile mid infrared stimuli (MIM) delivered directly on mice cortex or through a thinned skull increase both c-fos expression in cortical cells and neuronal activity. They also provided evidence that MIM delivered during an associative training task accelerates mouse learning. The work is very interesting and presents original results important for the neuroscience field interested in brain optical stimulation. However, sometimes it lacks detailed methodological information and some of the statistical analysis are not yet sufficiently robust to support authors' conclusions. Moreover, some assertions, such as that MIM effect is independent of the modification of tissue temperature, need to be supported by further investigation.

Major concerns:

c-fos experiments:

1-For the illumination duration of 20 s the authors calculate the proportion of cortical C-fos positive cells over the total cells (line 53), however they do not give any information about the method used to count the C-fos negative cells. Were the latter labeled by fluorescence dyes or counting was done on brightfield images?

2--The authors assessed the relation between c-Fos activation and illumination time by quantifying the total number of c-Fos positive cells in the different illuminating conditions (figure 1d). Since these measurements could be potentially affected by the cell density it would be more correct to

calculate the proportion of c-fos positive cell over the total number of cell as done for the condition 20s. Please mention the number of animals used in the different conditions.

3-Based on c-fos experiment the author conclude that “MIM application through opened skull could reliably induce neuronal activation”(lines 58-59). However, they did not provide evidences of the fact that positive cells are indeed neurons. c-fos can also be expressed in glial cells and astrocytes, as the same authors suggest when they consider that c-fos positives cell found in layer I are not neurons (lines 68-69) . A double labeling with a neuronal marker is required to determine the proportion of neurons among the c-fos labeled cells.

4-When discussing about the cellular mechanisms responsible of the effect produced by MIM the authors seems to exclude that it could be due to light induced increase of tissue temperature (lines 73-77). They justified their conclusion by citing two works (Owens et al 2019 and Ait Ouares et al 2019) on brain slices showing that light-induced increase of temperature reduces firing activity in all tested neuronal types. However, a precedent work had already shown an increase of firing activity of cortical neurons when stimulated by visible light in vivo condition, an effect that was proposed to be induced by temperature increase (Stujenske et al 2015; Cell Reports). Such apparent contradiction between the effect observed in vivo et in vitro has been explained by a stronger inhibitory effect of light on cortical interneurons than on pyramidal neurons (Owens et al 2019). Therefore, is still plausible that the effect observed in the present study is also due to temperature increase. To support the claim that MIM effect is temperature independent the authors should measure temperature in layer II/III before and during MIM stimulation, both using IR and visible light, and shown that: 1- temperature modification is below the limits reported to affect neuronal activity (ex. Ait Ouares et al figure 2) or 2- Temperature modification produced by visible light is equal or higher than that produced by IR.

Loose patch experiments:

5-Please provide evidences that the current deflections on electrophysiological traces (fig 1j) are real spikes and not an electrical artefacts produced by the MIM procedure. Does this activity can be blocked by TTX ?

6- Statistical analysis shown in figure 1J should be performed considering all recorded neurons (9) and not by selecting only the five on which un apparent effect of MIM is observed. Alternatively, a neuron by neuron analysis could be performed taking in account the MIM effect over the different trials. A supplementary figure showing the raster plot of all the recorded neurons would be also informative.

7-Please give information on how was calculated the normalized firing rate. A more direct method to evaluate light effect on firing would be to compare the absolute firing frequency in the 3 conditions (20 s before MIM, during MIM and 20s after). In any case a paired statistical test should be used.

Behavioral experiment:

8-The p values of figure 3d and 3e need to be corrected by thanking in account multiple testing. Since three statistical comparisons were made for each of the 6 sessions (Ctr Vs MIM thinned, Ctr vs MIM opened and MIM opened Vs MIM thinned) by using a Bonferroni correction no significant effect are observed. In order to make the statistical analysis more informative (does the data

support the absence of the MIM effect on success rate?) I suggest to performed Bayesian statistic (see Keysers et al Nature Neuroscience 2020).

9- Please detailed the method used to fit the learning curve.

Minor points:

10-Lines 20-22 “transcranial magnetic stimulation⁴ and transcranial direct-current stimulation⁵, have also been extensively practiced in healthy subjects with similar expectations” .Looking at the quoted references and the sentence that follow (lines 22-24), I suppose that “healthy subjects” should be replaced with “patients”.

11-Please give wavelength used for visual light stimulation

12-Could the author explain the reason to choose this quite atypical pattern of light stimulation (0.3 μ s duration @ 100KHz)?

13- The fact that continuous visual light has also been reported to increase cortical firing activity (Stujenske et al 2015) should be mentioned.

Nicola Kuczewski

We would like to express our deepest appreciation to the reviewers for their constructive comments and suggestions on our manuscript entitled “Non-invasive, opsin-free mid-infrared modulation activates cortical neurons and accelerates associative learning” (NCOMMS-20-35406-T). We completely agree with these comments and suggestions. In order to fully address their comments, we have performed extensive new experiments and new analysis that resulted in the following changes.

- (1) We carried out a systematic measurement of tissue temperature, mapping the spatial and temporal profiles of cortical tissue temperature in vivo under the mid-infrared light (MIR) stimulation as well as those under the visible light (VIS) control stimulation. We present the new data in Fig. 1i and Extended Data Fig. 1.
- (2) We performed a double labeling by using a neuronal marker NeuN together with c-Fos, and confirmed that the c-Fos positive cells emerged after MIR stimulation were dominantly neurons. We present a representative image in Fig. 1c and report the number in the main text.
- (3) As requested by the reviewer we improved the data reporting format in Fig. 1 and relevant texts, including: using c-Fos positive cell proportions instead of simple counts, and using absolute spike firing rate instead of normalized firing rate.
- (4) We performed additional loose-patch recordings and with tetrodotoxin (TTX) control experiments to confirm that the recorded spiking waveforms were neuronal firing signals but not optoelectrical artefacts. In addition, we performed the statistical analysis of all loose-patch recorded cells and also performed a neuron-by-neuron analysis over trials and the raster plot of all recorded cells. We present the new data in Fig. 1l, m and Extended Data Fig. 2, 3, 4.
- (5) We performed a new set of statistical analysis on the behavior data (Fig. 3) by using the Bayes Factor Hypothesis Testing as suggested by the reviewer. The results further confirmed our conclusion and are more informative, thus we also updated the main text accordingly.
- (6) We included the reference papers as suggested by the reviewers.

Reviewer's comments

Reviewer #1 (Remarks to the Author):

This paper uses a pulsed 5.6 μm laser to carry out optical stimulation in the mouse brain and in patch-clamped neurons.

1 The authors make a big deal of their study being different from traditional optogenetics approaches requiring the introduction of a light sensitive opsin into the neuronal cells. However in my opinion they mostly neglect a large body of work called "infrared neural stimulation". There is a good review of the subject

Chernov M, Roe AW. Infrared neural stimulation: a new stimulation tool for central nervous system applications. Neurophotonics. 2014;1(1):011011. doi:10.1117/1.NPh.1.1.011011

We thank the reviewer's suggestions that are essential for improving our manuscript, in particular to pointing out the important literature about general infrared neural stimulation. We have now cited and discussed this review paper and other related papers.

2 The wavelength they use (5.6 μm) may be novel but other wavelengths 980 nm, 1.45 μm , 1.87 μm , 2.12 μm have all been used. Absorption by water is what they all have in common and a small spot and a pulsed mode provide a confined thermal gradient to activate heat sensitive TRP ion channels

3 In fact TRPV4 was shown to be responsible in one study with 1.87 μm . Albert ES, Bec JM, Desmadryl G, et al. TRPV4 channels mediate the infrared laser-evoked response in sensory neurons. J Neurophysiol. 2012;107(12):3227-3234. doi:10.1152/jn.00424.2011

We thank the reviewer for mentioning the thermal effect induced by the infrared light in the wavelength range of 1-3 μm and the possible mechanism involving TRPV4 channels for this effect. However, our new experiment on measuring tissue temperature in vivo showed that the maximum temperature elevation was less than 2°C with the 5.6 μm light stimulation, which was much lower than the activation temperature measured in that reference paper (10 \pm 2 °C; Albert et al., J Neurophysiol, 2012) with the 1.87 μm light stimulation. In addition, two studies have found that the light illumination that is commonly used for optogenetics increases the temperature by \leq 2 °C (Owen et al., Nat Neurosci, 2019; Ait Ouares et al., Eur J Neurosci, 2019). This temperature elevation is associated with the inhibition of neuronal spiking in different brain areas and cannot explain the excitation effect observed in our experiments by the MIR (5.6 μm light) stimulation. We now mentioned this point in the main text. In addition, we also mention in the main text that the

wavelength of 5.6 μm is in the mid-infrared spectrum, as defined by the ISO20473 standard, and far from the near-infrared (NIR 0.78-3 μm wavelength) spectrum as previously used by others.

4 In addition to stating the average power of the laser (9mW) they should also state the peak power (300 mW)

We thank the reviewer for mentioning this key parameter. Following the suggestion, we now explain the peak power in detail in the methods section, as well as the rationale of why such a configuration was used (also as requested by reviewer #2).

Reviewer #2 (Remarks to the Author):

The authors show that optical pulsatile mid infrared stimuli (MIM) delivered directly on mice cortex or through a thinned skull increase both c-fos expression in cortical cells and neuronal activity. They also provided evidence that MIM delivered during an associative training task accelerates mouse learning. The work is very interesting and presents original results important for the neuroscience field interested in brain optical stimulation. However, sometimes it lacks detailed methodological information and some of the statistical analysis are not yet sufficiently robust to support the authors' conclusions. Moreover, some assertions, such as that MIM effect is independent of the modification of tissue temperature, need to be supported by further investigation.

We sincerely thank the reviewer for mentioning the importance of our study and for suggesting those important additional experiments. We completely agree with these suggestions and performed all the requested experiments, which greatly helped improving our manuscript.

Major concerns:

c-fos experiments:

1-For the illumination duration of 20 s the authors calculate the proportion of cortical C-fos positive cells over the total cells (line 53), however they do not give any information about the method used to count the C-fos negative cells. Were the latter labeled by fluorescence dyes or counting was done on brightfield images?

We thank the reviewer's suggestion. The c-fos negative cells were counted by the nucleus labelling by DAPI. We now mentioned it in the revised manuscript.

2--The authors assessed the relation between c-Fos activation and illumination time by quantifying the total number of c-Fos positive cells in the different illuminating conditions (figure 1d). Since these measurements could be potentially affected by the cell density it would be more correct to calculate the proportion of c-fos positive cell over the total number of cell as done for the condition 20s. Please mention the number of animals used in the different conditions.

We thank the reviewer's suggestions. We now calculate the c-fos positive cell proportion over the total number of cells (with DAPI staining) in Fig. 1e and stated the number of animals used in the different conditions in the main text.

3-Based on c-fos experiment the author conclude that “MIM application through opened skull could reliably induce neuronal activation” (lines 58-59). However, they did not provide evidences of the fact that positive cells are indeed neurons. c-fos can also be expressed in glial cells and astrocytes, as the same authors suggest when they consider that c-fos positives cell found in layer I are not neurons (lines 68-69). A double labeling with a neuronal marker is required to determine the proportion of neurons among the c-fos labeled cells.

We thank the reviewer’s suggestion for this important control experiment. We performed a new set of double labeling experiments with a neuronal marker (NeuN) together with c-Fos staining and found that 88.9 / 84.3 – 90.0 % (n = 11 slices from 4 animals) of c-Fos positive cells were neurons.

4-When discussing about the cellular mechanisms responsible of the effect produced by MIM, the authors seems to exclude that it could be due to light induced increase of tissue temperature (lines 73-77). They justified their conclusion by citing two works (Owens et al 2019 and Ait Ouares et al 2019) on brain slices showing that light-induced increase of temperature reduces firing activity in all tested neuronal types. However, a precedent work had already shown an increase of firing activity of cortical neurons when stimulated by visible light in vivo condition, un effect that was proposed to be induced by temperature increase (Stujenske et al 2015; Cell Reports). Such apparent contradiction between the effect observed in vivo et in vitro has been explained by a stronger inhibitory effect of light on cortical interneurons than on pyramidal neurons (Owens et al 2019). Therefore, is still plausible that the effect observed in the present study is also due to temperature increase. To support the claim that MIM effect is temperature independent the authors should measure temperature in layer II/III before and during MIM stimulation, both using IR and visible light, and shown that: 1- temperature modification is below the limits reported to affect neuronal activity (ex. Ait Ouares et al figure 2) or 2- Temperature modification produced by visible light is equal or higher than that produced by IR.

We thank the reviewer for suggesting this very important measurement. Following this suggestion, we used a miniature temperature probe (the same as that used in the study by Owen et al 2019) and measured the cortical tissue temperature at different sites in vivo under the same working conditions as in those neurophysiological experiments. The results are now shown in the Fig. 1i and Extended Data Fig. 1. In summary, we found that the maximum temperature elevation under either MIR (5.6 μm) or VIS light (594 nm) irradiation in our conditions was less than 2°C, which is in the same range of the temperature increase reported to reduce neuronal activity (Owen et al 2019; Ait Ouares et al 2019). In addition, the MIR light indeed induced a significantly lower temperature elevation than the

VIS light. Therefore, light-induced increase in tissue temperature (less than 2°C) cannot explain the excitation effect by MIR stimulation observed in our experiments. In addition, we also cite the paper mentioned by the reviewer (Ait Ouares et al 2019).

Loose patch experiments:

5-Please provide evidences that the current deflections on electrophysiological traces (fig 1j) are real spikes and not an electrical artefacts produced by the MIM procedure. Does this activity can be blocked by TTX ?

We thank the reviewer's suggestions. We performed additional loose-patch experiments that local TTX application blocked all the recorded spikes. We also analyzed the waveforms of the recorded signals to confirm that they were real neuronal spikes. We now present the new data in Extended Data Fig. 2 and 3.

6- Statistical analysis shown in figure 1J should be performed considering all recorded neurons (9) and not by selecting only the five on which unapparent effect of MIM is observed. Alternatively, a neuron by neuron analysis could be performed taking in account the MIM effect over the different trials. A supplementary figure showing the raster plot of all the recorded neurons would be also informative.

We thank the reviewer for this important suggestion. We now performed statistical analysis of all loose-patch recorded cells and also performed a neuron by neuron analysis over different trials. In addition, we show the raster plots of all recorded neurons. We present the new analyses in Fig. 1l-m and Extended Data Fig. 2-4.

7-Please give information on how was calculated the normalized firing rate. A more direct method to evaluate light effect on firing would be to compare the absolute firing frequency in the 3 conditions (20 s before MIM, during MIM and 20s after). In any case a paired statistical test should be used.

This requested analysis (Fig. 1m) is now performed with absolute firing rate and a paired statistical test (Wilcoxon signed-rank test).

Behavioral experiment:

8-The p values of figure 3d and 3e need to be corrected by thanking in account multiple testing. Since three statistical comparisons were made for each of the 6 sessions (Ctr Vs MIM thinned, Ctr vs MIM opened and MIM opened Vs MIM thinned) by using a Bonferroni correction no significant effect are

observed. In order to make the statistical analysis more informative (does the data support the absence of the MIM effect on success rate?) I suggest to performed Bayesian statistic (see Keysers et al Nature Neuroscience 2020).

Many thanks to this valuable suggestion. Following the suggestion, we applied the same Bayesian statistics as introduced in this important paper (Keysers et al, Nat Neurosci 2020) and obtained consistent results that more rigorously supported the original conclusion. The corresponding figure panels and texts have been updated with the new analysis results accordingly, and the main conclusion (that MIM application accelerates learning speed) still holds.

9- Please detailed the method used to fit the learning curve.

Added in the Methods.

Minor points:

10-Lines 20-22 “transcranial magnetic stimulation⁴ and transcranial direct-current stimulation⁵, have also been extensively practiced in healthy subjects with similar expectations”. Looking at the quoted references and the sentence that follow (lines 22-24), I suppose that “healthy subjects” should be replaced with “patients”.

Corrected.

11-Please give wavelength used for visual light stimulation

Provided.

12-Could the author explain the reason to choose this quite atypical pattern of light stimulation (0.3 μ s duration @ 100KHz)?

We appended the detailed explanation in the methods section. In short, this MIR pulsing laser was initially designed for the purpose of infrared spectroscopy; when we used it for this study, all the parameters were configured to maximize the total average output power at the 5.6 μ m wavelength.

13- The fact that continuous visual light has also been reported to increase cortical firing activity (Stujenske et al 2015) should be mentioned.

We thank the reviewer for mentioning this reference paper, which is now included.

Reviewer #1 (Remarks to the Author):

They appear to have failed to understand the difference between a highly localized increase in temperature which may only cover a region of a few microns or even nanometers, and which could not therefore produce any measurable temperature increase in the tissue because the heat is rapidly diffused. Saying the wavelength of 5.6 μm is different from 0.78-3 μm used in other studies is irrelevant. Both these wavelengths must be absorbed by water because there is no other chromophore that absorbs these wavelengths present at significant concentrations in cells or tissue. The most likely explanation is localized activation of heat-sensitive ion channels.

Reviewer #2 (Remarks to the Author):

The authors fully addressed my concerns.
I still have two minor points relative to the new modifications

1-There is an incongruence on the reported proportion of c-fos labeled cells, after 20 s MIM, between line 127: "(27.8 / 24.1 – 32.1 %, n = 21 slices from 7 mice)" and results presented on line 61: " 20 s': 23.3 / 19.8 – 31.5 %, n = 12 slices from 4 mice".

2-When giving the results of Bayesian statistic in the main text and in the legend of figure 3g please add the interpretation of the statistic " Evidence of absence, absence of evidence, evidence of presence or strong evidence of presence", using the criteria described in the methods. In figure 3g please add BF10= before numbers.

Best regards,
Nicola Kuczewski

We would like to express our deepest appreciation to the reviewers for their constructive comments and suggestions on our revised manuscript entitled “Non-invasive, opsin-free mid-infrared modulation activates cortical neurons and accelerates associative learning” (NCOMMS-20-35406A). The new comments have been fully addressed in the revised version; please see below a point-by-point response.

REVIEWER COMMENTS

Reviewer #1 (Remarks to the Author):

They appear to have failed to understand the difference between a highly localized increase in temperature which may only cover a region of a few microns or even nanometers, and which could not therefore produce any measurable temperature increase in the tissue because the heat is rapidly diffused. Saying the wavelength of 5.6 μm is different from 0.78-3 μm used in other studies is irrelevant. Both these wavelengths must be absorbed by water because there is no other chromophore that absorbs these wavelengths present at significant concentrations in cells or tissue. The most likely explanation is localized activation of heat-sensitive ion channels.

We apologize for having not properly interpreted the temperature measurements and thank the reviewer for this clear suggestion, which has been now included in the new manuscript (please see the discussion text following the temperature measurement data description).

Reviewer #2 (Remarks to the Author):

The authors fully addressed my concerns.

I still have two minor points relative to the new modifications

1-There is an incongruence on the reported proportion of c-fos labeled cells, after 20 s MIM, between line 127: “(27.8 / 24.1 – 32.1 %, n = 21 slices from 7 mice)” and results presented on line 61: “ 20 s’: 23.3 / 19.8 – 31.5 %, n = 12 slices from 4 mice ”.

We would like to thank the reviewer very much for careful inspection of our data. In fact, these are two independent sets of experiments (though performed under the same condition). A rank sum test for these two data showed $p = 0.93$, and thus these results were not significantly different. In the revised

manuscript, we wrote 'a new set of experiments' before the description of the second set of experiments.

2-When giving the results of Bayesian statistic in the main text and in the legend of figure 3g please add the interpretation of the statistic " Evidence of absence, absence of evidence, evidence of presence or strong evidence of presence", using the criteria described in the methods. In figure 3g please add BF10= before numbers.

Done.